# Synthesis, Characterization, and Evaluation of the Antimicrobial Effects and Cytotoxicity of a Novel Nanocomposite Based on Polyamide 6 and Trimetaphosphate Nanoparticles Decorated with Silver Nanoparticles

**DOI:** 10.3390/antibiotics13040340

**Published:** 2024-04-08

**Authors:** Leonardo Antônio de Morais, Francisco Nunes de Souza Neto, Thayse Yumi Hosida, Danilo Martins dos Santos, Bianca Carvalho de Almeida, Elisabete Frollini, Sergio Paulo Campana Filho, Debora de Barros Barbosa, Emerson Rodrigues de Camargo, Alberto Carlos Botazzo Delbem

**Affiliations:** 1Department of Preventive and Restorative Dentistry, School of Dentistry, São Paulo State University (UNESP), Rua José Bonifácio, 1193, Araçatuba 16015-050, São Paulo, Brazil; leonardo.a.morais@unesp.br (L.A.d.M.); francisco_nsn@yahoo.com.br (F.N.d.S.N.); thayse.hosida@unesp.br (T.Y.H.); bianca.c.almeida@unesp.br (B.C.d.A.); debora@foa.unesp.br (D.d.B.B.); 2Sao Carlos Institute of Chemistry, University of Sao Paulo, Av. Trabalhador Sao-Carlense, 400, São Carlos 13566-590, São Paulo, Brazil; martinsdanilo.9@gmail.com (D.M.d.S.); frollini@iqsc.usp.br (E.F.); campana@iqsc.usp.br (S.P.C.F.); 3Center for Exact Sciences and Technology, Federal University of São Carlos (UFSCAR), Av. Trab. São Carlense, 400, São Carlos 13566-590, São Paulo, Brazil; camargo@ufscar.br

**Keywords:** phosphates, silver, nanoparticles, biocompatible materials, polymers

## Abstract

This study aimed to develop a polymeric matrix of polyamide-6 (P6) impregnated with trimetaphosphate (TMP) nanoparticles and silver nanoparticles (AgNPs), and to evaluate its antimicrobial activity, surface free energy, TMP and Ag^+^ release, and cytotoxicity for use as a support in dental tissue. The data were subjected to statistical analysis (*p* < 0.05). P6 can be incorporated into TMP without altering its properties. In the first three hours, Ag^+^ was released for all groups decorated with AgNPs, and for TMP, the release only occurred for the P6-TMP-5% and P6-TMP-10% groups. In the inhibition zones, the AgNPs showed activity against both microorganisms. The P6-TMP-2.5%-Ag and P6-TMP-5%-Ag groups with AgNPs showed a greater reduction in CFU for *S. mutans*. For *C. albicans*, all groups showed a reduction in CFU. The P6-TMP groups showed higher cell viability, regardless of time (*p* < 0.05). The developed P6 polymeric matrix impregnated with TMP and AgNPs demonstrated promising antimicrobial properties against the tested microorganisms, making it a potential material for applications in scaffolds in dental tissues.

## 1. Introduction

The loss of tissue resulting from dental caries or trauma underscores the importance of research in tissue engineering. This field focuses on developing biomaterials aimed at regenerating tissue structure and restoring both form and function [1]. Over the last decades, polymeric materials in the form of three-dimensional porous substrates (scaffolds) have been used to act as an extracellular matrix in the growth, proliferation, and support of the formation of new living tissues [2,3].

As an alternative to scaffolds, polymeric fibers of polyamide-6 (P6) are known to be excellent biomaterials due to the high relation between their surface area and volume, the interconnectivity of their fibers, and the existence of interstitial space, presenting different applications in the biomedical field [4,5]. However, the characteristics of the surfaces such as hydrophilicity and low functionality of P6 make it difficult to apply as a scaffold [6]. To minimize or resolve this issue, the functionalization of P6 with phosphates has been used to onset the in vitro mineralization of bone and dental tissues [7].

Anionic groups, such as phosphates, have the ability to form apatite crystals [8], which can also induce the mineralization of dental tissue [9]. In this case, phosphates react with the amino or hydroxyl groups of proteins, attaching them to amino acid chains [10]. Sodium trimetaphosphate (TMP) is the most studied cyclophosphate for dental applications [11] due to its anticaries action [12,13]. Favretto et al. [13] showed that the treatment of dentin with TMP leads to the precipitation of apatite on the surface and into the dentinal tubules, increasing the mineral content of the dentin. The use of phosphates promotes a greater number of electron donor sites to the substrate surface, leading to the higher adsorption of calcium phosphate on the surface [14]. Furthermore, cell adhesion is influenced by electrostatic interaction with the surface of biomaterials and is related to the polar components of surface free energy as higher values of electron donor sites lead to greater cell adhesion [15]. 

P6 and TMP do not present antimicrobial activity [16,17], but when TMP is impregnated with silver nanoparticles (AgNPs), this compound demonstrates significant antimicrobial activity against *Streptococcus mutans* and *Candida albicans*, as demonstrated by Mendes-Gouvêa et al. [17] in their study with solutions of these compounds. Silver nanoparticles have stability [17], low toxicity, and antimicrobial activity against *Streptococcus mutans* and *Candida albicans* [17]. Therefore, it would be interesting to develop a new material that presents antimicrobial action and low cellular cytotoxicity to be used as a scaffold for the regeneration of dental tissues based on P6 and TMP impregnated with AgNPs.

The purpose of this study was to develop and characterize a material formed by a polymeric matrix of P6 impregnated with TMP nanoparticles in different concentrations decorated with AgNPs, and to evaluate its antimicrobial activity, surface free energy, TMP and Ag^+^ release, and cellular cytotoxicity.

## 2. Results

### 2.1. Determination of Minimum Inhibitory Concentration (MIC)

For planktonic cells of *C. albicans*, the minimum inhibitory concentration (MIC) and minimum fungicidal concentration (MFC) values of AgNPs were 9.40 and 300.9 mg/mL with or without the presence of NH_3_ (Table 1). For *S. mutans*, the MIC and minimum bactericidal concentration (MBC) values were 601.9 and 601.9 mg/mL, respectively, for AgNPs with NH_3_ and 300.9 and 300.9 mg/mL for AgNPs without NH_3_ (Table 1).

### 2.2. Synthesis and Characterization of Silver and TMP Nanoparticles

The AgNPs were processed and characterized as described by Neto et al. [18]. Briefly, AgNPs showed UV-Vis spectra of AgNP colloids in the range of 300 nm–800 nm, confirming their formation. A macroscopic characteristic of AgNPs is their yellow color, characteristic of a spherical morphology, as shown by Neto et al. [18]. Furthermore, the absorption spectrum of AgNPs showed a symmetric SPR band at 410 nm, confirming that Ag^+^ ions were reduced to Ag0 in the alcoholic medium. The methodology used resulted in nanometer-scale particles, without mixing crystalline phases.

To evaluate the effect of the ball-milling process on the morphology and size of TMP nanoparticles, SEM and TEM measurements were performed. Figure 1A shows that before the ball-milling process, the TMP particles were micrometer-sized and highly agglomerated. After this process (Figure 1B), the particles were dispersed with a spherical morphology and a size of 73.4 ± 10.4 nm (Figure 1C), showing that this methodology proved to be efficient in reducing the size of TMP particles.

Figure 2A shows the X-ray diffractogram of TMP samples after 48 h submitted to milling processing. The obtaining TMP, whose empirical formula is (Na_3_P_3_O_9_), was evidenced by comparing the reflection patterns obtained experimentally and the corresponding crystallographic record (PDF#72-1628). The solid crystallized in an orthorhombic lattice (the unit cell has three axes with different edge lengths, forming right angles).

The analysis of the survey spectrum (low-resolution spectrum), Figure 2B, shows the presence of the following analytes on the surface of the material ground for 48 h: Si, P, C, O, Fe, and Na. Figure 2C shows that the phosphor signal (P 2p) was deconvoluted into two components. The main peak centered at approximately 133 eV (P 2p3/2) indicates that the oxidation state of P in the TMP ground for 48 h is pentavalent, with the P atom forming four chemical bonds (tetracoordinated P), as is the case with cyclic phosphates. The second peak at approximately 135 eV (P 2p1/2) is attributed to the P-O-P binding of metaphosphates. The oxygen signal (O 1s) is deconvoluted into three components (Figure 2D). The O 1s peaks indicate the presence of different chemical states of the oxygen atom. The peak of O 1s close to 530 eV is attributed to oxygen bound to the phosphate group (P=O). The peak close to 532 eV is attributed to the P-O-P grouping. The peak close to 535 eV is attributed to the chemisorbed O atom coming from water molecules.

### 2.3. Synthesis and Characterization of P6-TMP-AgNP Membranes

The electrospinning process of the membranes produced homogeneous fibers, without porosities or surface defects (Figure 3) and the presence of AgNPs decorating the membrane (Figure 4). The mean (±SD) diameter of the fibers was 140 ± 37 nm for P6, 177 ± 25 nm for P6-TMP-2.5%, 223 ± 39 nm for P6-TMP-5%, and 433 ± 74 nm for P6-TMP-10%. The chemical elements sodium (Na), phosphorus (P), and silver (Ag) were detected by EDX analysis (Figure 5). Absorption spectroscopy in the infrared region showed the presence of characteristic bands of the CO, NH, and CH bonds of P6, as well as a 1100 cm^−1^ band, characteristic of symmetric stretching (POP) in phosphates, showing the incorporation of TMP in the P6 fibers (Figure 6). For AgNP membranes, it was not possible to observe the aforementioned absorption bands.

The structural characterization of the polymeric material and its nanocompounds (NMR technique) showed that the materials have five chemical shifts that are attributed to the functional groups of P6 (Figure 7A) and that there is no shift in the nanocompound peaks compared to P6; this means that the TMP is physically connected to P6. The 31P-NMR spectrum of TMP shows two characteristic peaks at −15.7 and −19.2 ppm (Figure 7B). The peak centered at −15.7 ppm is attributed to the phosphorus (PA) site and the peak centered at −19.2 ppm to the two atoms (PB).

The thermogravimetric curves of the samples of P6 and its nanocompounds are shown in Figure 8A. The thermogravimetric curves showed that the evolution of mass loss is slower in membranes due to the presence of nanoparticles, indicating that although TMP nanoparticles promote an increase in mass loss, the membrane degradation kinetics are slower. Figure 8B shows that P6-TMP showed an increase in glass transition temperature (*Tg*) compared to P6, as there was an increase in the viscosity of polymer solutions leading to a decrease in free volume, thus restricting the mobility of the polymer chain and increasing the value of *Tg*.

The incorporation of TMP in the polymeric matrix of P6 leads to an improvement in the mechanical properties of the polymeric matrix, including an increase in the modulus of elasticity and tensile strength. The membrane with P6-TMP-2.5% showed the best results for elasticity and traction (Figure 9).

### 2.4. Surface Free Energy of Membranes

For the use of materials in tissue engineering, the scaffold substrate is of great importance for the formation of a biocompatible biomaterial, as it allows for the growth, migration, and differentiation of cells on its surface, making it considered a biocompatible material. In this context, contact angle measurements become important. Table 2 and Table 3 show the results of the surface free energy for P6 and its nanocomposites without and with decoration with Ag nanoparticles.

### 2.5. Release of Ag^+^ and TMP

The greatest amount of Ag^+^ was released in the first three hours for all groups decorated with AgNPs (Figure 10A) and the mean release did not differ between these groups (*p* > 0.05). The values accumulated during 24 h (Figure 10B) were higher for the P6-Ag and P6-TMP-5%-Ag membranes than for the other groups (*p* < 0.001). There was no release for groups without AgNPs.

There was a small release of TMP in the first three hours for groups P6-TMP-5% and P6-TMP-10% (Figure 10C), and for the other membranes, no releases were detected. The mean or accumulated release indicates that there was no TMP release, regardless of the group (Figure 10C,D).

### 2.6. Evaluation of the Antimicrobial Activity of Membranes

In the agar diffusion assay, the AgNP-containing membrane groups showed the formation of the inhibition halo for *C. albicans* and *S. mutans* (Figure 11). For *C. albicans*, all groups containing AgNPs showed an inhibitory action. The groups P6-TMP-5%-Ag and P6-TMP-10%-Ag did not differ from each other (*p* > 0.05), although they presented the largest inhibition zones when compared with the other groups (*p* < 0.001). For *S. mutans*, the groups P6-TMP-2.5%-Ag, P6-TMP-5%-Ag, and P6-TMP-10%-Ag did not differ in the formation of the inhibition halo (*p* > 0.05); however, these were higher compared with the other groups (*p* < 0.001). The groups without AgNPs presented no antimicrobial activity, while the positive control (chlorhexidine) presented the greatest inhibition halos (*p* < 0.001).

The groups P6-TMP-2.5%-Ag and P6-TMP-5%-Ag with AgNPs showed a greater reduction in CFU for *S. mutans* in comparison to the other groups (*p* < 0.001), with a greater reduction in the time of 18 h. For *C. albicans*, all groups showed a reduction in CFU in comparison to the control, with no statistical difference between them. For *C. albicans*, the times of 1 and 2 h showed a greater reduction in CFU for all groups evaluated (Figure 12).

### 2.7. Evaluation of Cytotoxicity by MTT Assay

Groups with P6-TMP showed greater viability when compared to groups with AgNPs, regardless of time (*p* < 0.05). Groups with AgNPs showed reduced cell viability for all groups, with a greater reduction at 24 h when compared to 48 and 72 h (*p* < 0.05). All groups showed an increase in viability within 72 h (*p* < 0.05) (Figure 13).

## 3. Discussion

Technological advancements have been driving the research and development of new biomaterials for the treatment and recovery of injuries resulting from trauma and degenerative diseases [1,19,20]. In dentistry, the focus has been on dental caries control [21]. An ideal membrane must meet specific criteria before being used for clinical applications, such as biocompatibility and biodegradability, as well as an appropriate surface and a porous structure that allows for cell migration, attachment, infiltration, proliferation, and differentiation. Additionally, the membrane needs to possess sufficient mechanical strength to support tissue regeneration while maintaining space [22]. Currently, a variety of biomaterials, including synthetic and natural materials, are available on the market for the production of barrier membranes. With the selection of the right biomaterial, it is crucial to use appropriate manufacturing methods when developing a membrane with good properties and the ability to incorporate other composites with therapeutic capabilities [23,24]. The present study synthesized a P6 membrane impregnated with TMP nanoparticles decorated with AgNPs, demonstrating significant antimicrobial action against *Streptococcus mutans* and *Candida albicans*.

The antimicrobial activity of AgNPs is well established in the literature; however, their mechanism of action is still unclear. When crossing the cell membrane, AgNPs bind and adhere to it, causing damage to the cell [8], or, when permeating the cell membrane, they dissolve and the AgNP ions are released [17]. However, when AgNPs penetrate the microorganism’s cell, they damage the DNA, essentially through the creation of ROS (reactive oxygen species) [25] and, due to the respiratory process, they can alter the electrochemical gradient of protons, interrupting the process of cell ATP (adenosine triphosphate) synthesis, leading to cell death. Furthermore, the combination of AgNPs and other compounds can improve its antimicrobial activity [26].

Among metallic nanoparticles, AgNPs have been specifically studied as an alternative antimicrobial agent in the control of oral biofilms against a wide spectrum of microorganism species [27]. Ammonia (NH_3_) is among the most commonly used stabilizing agents in the chemical reaction method to synthesize AgNPs, as it can control the growth of particles and prevent their aggregation. Therefore, through NH_3_, it stimulates soluble silver complexes, which stabilize the AgNPs that retain the free silver ions responsible for increasing particle size [28]. 

Despite this characteristic, in the minimum inhibitory concentration test, a lower concentration of AgNP solution without NH_3_ was needed to inhibit the growth of *S. mutans* when compared to the solution with NH_3_. Furthermore, the presence of NH_3_ as a stabilizing agent during the synthesis of nanoparticles ended up interfering with their color as well. Yunji Lee et al. [29] reported that as they added ammonia to the silver synthesis, the solution showed a darker color by visual observation.

*C. albicans* planktonic cells are more susceptible than *S. mutans* planktonic cells due to their structural differences in cell membranes. Bacterial membranes are negatively charged due to anionic phospholipids, while in fungal membranes, the charge is similar to that in neutral and rigid eukaryotic membranes [30]. Abbaszadegan et al. [31] state that the stabilizers and agents used influence the antimicrobial activity of AgNPs, as they depend on the external surface charge of the particles during synthesis. Thus, in this work, the synthesized AgNPs had a negative surface charge, as in the Zeta-potential electrochemical test, it was observed that the AgNPs showed negative values (−30 mV). Mandal et al. [32], in their study, report that the external charge of bacterial membranes influences the mechanism of action of AgNPs, and this may explain the decreased susceptibility of *S. mutans* and the need for a higher concentration to inhibit its growth, as seen in Table 1.

According to the synthesis and characterization of the membranes, this showed promising results, since the fibers were homogeneous, without porosity and surface defects, in addition to the incorporation of AgNPs and TMP. A micro- and nano-scale topographical pattern decreases bacterial adhesion and bactericidal effects [33]. Appropriate mechanical properties are important parameters in the production of scaffolds to be used in tissue engineering. One of the parameters for scaffold applications is that they must be elastic enough to resist pressure and flow during long-term implantation without tearing. The incorporation of TMP in the polymeric matrix of P6 leads to an improvement in the mechanical properties of the polymeric matrix, including an increase in the modulus of elasticity and the tensile strength (which were increased in the composite with 2.5% of TMP).

The thermogravimetric curves showed that, chemically, the increase in the thermal stability of P6 with the insertion of TMP is due to the formation of hydrogen bonds between the amide groups of P6 and the oxygen atom of TMP, with a higher energy requirement for the rupture of such links. The addition of TMP leads to a change in the initial degradation temperature due to its good dispersion in the polymer matrix, allowing the nanoparticles to fill the spaces between the polymer chains, resulting in the need for a higher temperature to decompose the polymer matrix of P6.

A possible explanation for the increase in membrane *Tg* is that there is an increase in the interactions between chains through hydrogen interactions, and this collective increase in rigidity contributes to the increase in *Tg*. After the insertion of the TMP, the polymeric chains interact through hydrogen bonds, and for the movement of the molecular segment of the membranes, greater thermal energy is required.

For the evaluation of surface free energy, in all nanocomposites, the solid surface energy (γ_s_) was lower than that of P6. The water droplet penetrated the solid and filled a portion of the pores, forming a surface that belongs to both the solid and the liquid (Table 2). The water droplet interacted more with the nanocomposites compared to P6 (Table 3), as TMP is a compound that contains P and O atoms, which have a large number of electrons (Lewis bases) that interact with the electron-deficient atoms (Lewis acids) present in P6, thus forming a hydrogen bond, which characterizes the surface of the nanocomposites as hydrophilic [34]. The data from Δ*G*_sws_ corroborates these results, showing positive values, indicating that the membrane impregnated with nanoparticulate TMP led to a hydrophilic surface. However, the increase in TMP concentration made the membrane surface less hydrophilic. A detailed analysis of the SEM image of the P6 nanocomposite impregnated with 10% TMP (Figure 3D) showed the presence of fibers with a very small width, known as the spider-net. The formation of the spider-net structure can be explained by the formation of hydrogen bonds between the main polyamide chain of its amide groups (CO-NH), oligomers, and monomeric ionic species (-CONH_2_-^+^) [35]. The presence of oligomers and monomeric ions in solution is due to formic acid (a monoprotic solvent with a high dielectric constant), which is capable of attacking the lactam and producing a series of short oligomers and monomers [36]. Increasing the concentration of TMP seems to intensify this reaction and spider-net formation, leaving the surface less hydrophilic. This is because the cyclophosphates, including TMP, are electrolyte, and the electrical conductivity of polymer solutions increases with their insertion. The high electrostatic charge leads to a strong electrostatic attraction between polymer molecules and the TMP in its dissociated form. Likewise, this led to a larger fiber diameter (Figure 3D), showing that, with the insertion of 10% TMP, the average fiber diameter tended to increase, ranging from 140 nm (P6 without TMP) to 433 nm. Therefore, in the P6 polymer solutions with TMP, there was greater charge mobility, which, associated with the external electric field applied, caused the droplet to be more elongated, spreading into thinner segments and resulting in fibers with larger diameters and homogeneous morphology.

Increasing hydrophilicity is important for biomaterials as it enhances cell adhesion and migration into the scaffold [37,38]. The presence of TMP in the P6 fibers can act as nucleation sites for apatite precipitation, inducing cell adhesion and the formation of hard tissue, which is important for bone repair. However, the hydrophobicity achieved in this study may also favor bacterial adhesion, as the insertion of TMP into P6 did not produce a superhydrophilic surface. A surface is characterized as superhydrophilic when the contact angle is <10° and leads to a reduction in microbial adhesion on material surfaces [38]. In this study, the average value (±SD) of the water contact angle was P6 = 63.0° ± 5.2 and P6 + TMP = 45.6° ± 2.8 (data not described in the results). Thus, the addition of an antimicrobial agent, such as nanoparticulate silver, was essential to obtaining an antibiofilm material.

It is already known in the literature that TMP has no effect on the microbiological composition of dental biofilm [21]. In this study, it was possible to demonstrate that TMP with AgNPs did not prevent the antimicrobial effect of silver; in fact, the property of polyphosphate adsorption on protein by hydroxyl and amino sites [39,40] can increase the availability of Ag^+^ and improve the antimicrobial effect of these compounds. This was observed in this study, especially for *C. albicans*, in which the groups with the highest concentration of TMP (5% and 10%) decorated with AgNPs showed a greater halo of inhibition in comparison to the other groups tested, with no statistical difference between them. As for *S. mutans*, the 2.5%, 5%, and 10% TMP groups associated with AgNPs showed better performance when compared to P6-Ag (Figure 8). These results demonstrate that not only was the presence of AgNPs relevant to the antimicrobial efficacy of membranes, but their association with TMP had a significant role in increasing the antimicrobial potential. In their study, Humphreys et al. [41] demonstrated that the synergism between polyphosphates and Ag^+^ can increase the outer membrane permeability of bacteria associated with the polyphosphate sequestration of divalent cations [42,43,44], allowing silver to cross the membrane, causing damage to cells, such as protein inactivation and loss of DNA replication [45].

According to Mendes-Gouvêa et al. [18], the groups with the highest concentration of TMP associated with AgNPs were the groups that obtained greater antimicrobial action compared to other groups with lower concentrations. The same pattern can be observed in the halo of the inhibition test for *C. albicans*, in which the groups of P6-5%-TMP-Ag and P6-10%-TMP-Ag associated with AgNPs showed a greater halo of inhibition in comparison to the other groups. However, when analyzing the CFU data, the highest concentration of TMP did not potentiate the antimicrobial action of AgNPs for *S. mutans* and *C. albicans.*

Furthermore, in the first three hours, the greatest release of Ag^+^ occurred for all groups decorated with AgNPs, and for TMP, a small release in the period of 10 h for all groups decorated with AgNPs. Therefore, the greater release of Ag^+^ in the first three hours may have influenced the greater reduction in CFU for *C. albicans* in groups P6-Ag-5%TMP and P6-Ag-10%TMP in this period. Importantly, this result may be related to the fact that biofilms of *Candida* spp. species may exhibit up to 65 times more tolerance to death by metals than the corresponding planktonic cultures [46]. Furthermore, it would be interesting to evaluate their action over longer periods of formation, as AgNPs can reduce their effectiveness, as they tend to form aggregates in the absence of support. Therefore, substrates decorated with AgNPs present increased antimicrobial activity for longer periods [47].

Regarding *S. mutans*, the greatest CFU reduction for groups containing AgNPs was within 18 h, which may be related to a small release of TMP within 10 h, which would potentiate the action of AgNPs, as TMP has a large affinity for metal ions (Mg^2+^, Ca^2+^, K^+^, Al^+^, Fe^3+^), forming ionic complexes [48]. This property allows this phosphate to bind to the cell wall of microorganisms, thus increasing cell permeability, allowing silver to pass through the membrane and then cause cell death. Due to its cellular structure being more complex than that of bacteria due to its yeasts and hyphae, the results were different for *C. albicans*, which probably hindered the binding effects of TMP and increased permeability.

The groups with AgNPs showed reduced cell viability for all groups, with a greater reduction over 24 h compared to 48 and 72 h. After 24 h, there was an increase in cell viability. The decrease in the reduction of MTT to formazan within 24 h shows the greater sensitivity of cells to AgNPs. However, in this work, it was possible to notice an increase in the remaining viable cells at 48 and 72 h, suggesting that the proliferative capacity was preserved in the groups with P6-TMP. Even so, in 72 h, the cell viability increased for all groups, with no statistical difference between them. Even the genotoxic and cytotoxic effects that are produced by nanocompounds that are in direct contact with cells can be clinically attenuated or neutralized, acting as a physical barrier depending on thickness or even the interaction of AgNPs with the components of the medium, which can interfere with the results. Dutra-Correa et al. [49] demonstrated in their study with AgNPs associated with adhesives that there was also an increase in cell viability within 72 h, which could have been caused by the physical barrier of the adhesive with the tested cells, leading to the possible neutralization of the cytotoxic effect of AgNPs.

Although the antimicrobial results are promising, more studies are still needed to better evaluate the remineralizing and antimicrobial action of these nanocompounds, such as the use of tests on mature biofilms, cytotoxicity tests, and others.

## 4. Materials and Methods

### 4.1. Synthesis of AgNP and TMP Nanoparticles

The AgNPs were processed and characterized as described by Neto et al. [19], and TMP nanoparticles were obtained using a solid-state reaction methodology (milling) described by Danelon et al. [50]. To prepare nanosized TMP, 0.23 mol of commercial sodium trimetaphosphate (Na_3_O_9_P_3_, Aldrich, St. Louis, MO, USA, 95% purity CAS 7785-84-4) was ball-milled using 500 g of zirconia spheres (2 mm diameter) in 1 L of isopropanol for 48 h. After 48 h, the resulting powder was separated from the alcoholic medium by filtration, dried at 85 °C for 12 h, and ground in a mortar.

### 4.2. Determination of the Minimum Inhibitory Concentration (MIC) of AgNPs

The Clinical Laboratory Standards Institute has defined guidelines for carrying out the microdilution method (CLSI: M27-A2 and M07-A9). AgNPs were synthesized in two solutions, one with the presence of NH_3_ and the other without NH_3_. In geometric progression, the AgNPs were diluted in deionized water at a dilution of 2 to 1024 times. Subsequently, each concentration of AgNPs obtained previously was diluted in RPMI 1640 medium (Sigma-Aldrich, St. Louis, MO, USA) for *Candida albicans* (ATCC 10231) and in Brain–Heart Infusion (BHI, Difco, Le Pont de Claix, France) for *Streptococcus mutans* (ATCC 25175) in (1:5). The 24 h culture inocula were adjusted in saline (0.85% NaCl) to standard turbidity similar to 0.5 McFarland. The suspensions of each microorganism were diluted (1:5) in saline solution and then diluted for *C. albicans* in RPMI 1640 (1:20) and for *S. mutans* or BHI broth. Each microorganism suspension was inserted into the wells of microtiter plates (100 μL) containing 100 μL of each concentration of nanocompounds. After that, they were incubated at 37 °C in the incubator. Visually, the MICs were determined after 48 h with the lowest concentration of AgNPs without the growth of microorganisms [17]. After 48 h in 5% CO_2_ at 37 °C, each well was plated on SDA (for *C. albicans*) or BHI (for *S. mutans*) agar to determine the Minimum Bactericidal Concentration (MBC) and Minimum Fungicide Concentration (MFC) of the solutions against the tested strains. The assays were carried out at three different times and in triplicate. 

### 4.3. Preparation of P6 Membranes Containing TMP and Decorated with AgNPs

The production of P6 nanofibers (Sigma-Aldrich 99%) was carried out following the methodology proposed by Andre et al. [51] and Mi et al. [52] with modifications. P6 was solubilized in formic acid at a concentration of 15% (*m*/*v*) under constant magnetic stirring for 5 h at 25 °C. P6 containing TMP (2.5%, 5% and 10%) was solubilized in formic acid as previously described. Then, the homogenized solution was electrospun in an electrospinning device (EC-DIG—IME Technologies, Geldrop, Netherlands) with a voltage of 25 kV and a flow rate of 10 μL.h^−1^ using a collecting substrate positioned at a distance of 10 cm from the syringe. Then, the P6-TMP membranes, at concentrations of 0%, 2.5%, 5%, and 10%, were covered with AgNPs by immersing the membranes in 15 mL of an AgNP suspension for 24 h. Based on previous pilot studies, the concentrations (P6, P6-TMP-2.5%, P6-TMP-5%, and P6-TMP-10%) showed the best mechanical and chemical properties. Because of this, these were the elective concentrations for the experimental tests.

### 4.4. Characterization of TMP, AgNP, P6, P6-TMP, and P6-TMP-AgNP

TMP, AgNP, and P6-TMP-AgNP membranes were characterized in the Interdisciplinary Laboratory of Electrochemistry and Ceramics at the Department of Chemistry of the Federal University of São Carlos (UFSCar). The different concentrations of the membranes obtained were characterized by the technique of energy-dispersive X-ray spectroscopy (EDX) [53], absorption spectroscopy in the mid-infrared region (FTIR) [54], scanning electron microscopy (SEM) [17], solid-state nuclear magnetic resonance (ss-NMR) [55], thermogravimetric analysis (TGA) [56], differential scanning calorimetry (DSC) [57] and thermodynamic and mechanical analysis (DMTA) [58], UV-Vis absorption spectrophotometry [59], X-ray Diffraction (DRX) and Energy Conversion Solutions (XPS) [60], Transmission Electron Microscopy (TEM), and dynamic light-scattering (DLS) [61]. The data for elasticity modulus, elongation at break, and tensile strength were normally distributed and were subjected to one-way ANOVA, followed by the Student–Newman–Keuls post hoc test (*p* < 0.05; SigmaPlot 12.0 software, Systat Software Inc., San Jose, CA, USA).

### 4.5. Surface Free Energy of Membranes

The substrate used for the application of P6-TMP-Ag nanofibers was a double-sided foam adhesive tape (Scotch, 3M, São José do Rio Preto, São Paulo, Brazil, 12 mm × 5 m) and a glass slide as a base for film formation. The surface free energy (γ_s_) and its components, apolar (γ_s_^LW^: Lifshitz van der Waals or dispersive energy) and polar (γ_s_^AB^: acid/base), were determined on the surfaces of the nanofibers by measuring the contact angle of probe liquids. For contact angle measurement, the nanofibers were kept in an environment at 23.3 °C (±0.3) for 45 min to obtain stability of the formed film [62].

Measurements were performed using an automatic goniometer (DSA 100S, Krüss, Hamburg, Germany) and three probe liquids with known surface energy parameters: water (polar), diiodomethane (apolar; Sigma-Aldrich Co., St. Louis, MO, USA), and ethylene glycol (polar with acid and base components; Sigma-Aldrich Co., St. Louis, MO, USA).

For contact angle determination (θ°), a volume of 0.3 μL of each liquid was automatically dispensed onto the surface (6 mm × 6 mm) of the nanofibers using a glass syringe (500 μL) and a 0.5 mm gauge needle [63]. After 1 s, the contact angles of the droplets (left and right) were measured using a CCD camera for image capture and the Tangent method (Drop Shape Analysis DSA4 Software, version 2.0-01, Krüss). Each droplet was measured 6 times for 2 s at a temperature of 23.3 °C (±0.3) [62] and relative humidity of approximately 38.1% (±3.3). The parameters γ_s_^LW^ and γ_s_^AB^, and the acid (γ_S_^+^, receptor component) and base (γ_s_^−^, donor component) components of the surface free energy (mN/m), were calculated according to the van Oss, Chaudhery, and Good model [64,65,66,67] for surface free energy determination [68,69]. The total interaction free energy (Δ*G*_sws_^Total^) between the nanofibers and water was also calculated to determine the hydrophobicity/hydrophilicity of the nanocomposites. When Δ*G*_sws_ > 0, the surface was considered hydrophilic, and if Δ*G*_sws_ < 0, the surface was considered hydrophobic: Δ*G*_sws_^Total^ = −2(√γ_s_^LW^ − √γ_w_^LW^)^^2^ − 4(√γ_s_^+^γ_s_^−^ + √γ_w_^+^γ_w_^−^ − √γ_s_^+^γ_w_^−^ − √γ_s_^−^γ_w_^+^) [15,65,66,67,68,69]. The data were normally distributed and were subjected to two-way ANOVA (with or without silver and groups as variation factors), followed by the Fisher’s LSD post hoc test (*p* < 0.05; SigmaPlot 12.0 software).

### 4.6. Release of TMP and Ag^+^ from P6-TMP-AgNP Membranes

Six disks from each group were made with the aid of a 5 mm diameter perforator. These samples were placed in centrifuge microtubes containing 2 mL of deionized water and kept in an oven at 37 °C. After a period of 1, 2, 3, 4, 5, 10, 12, 14, 16, 18, 20, and 24 h, the dosages of TMP and Ag^+^ released through the cumulative method were performed. The TMP released by the membranes was quantified by the determination of phosphorus, after hydrolysis in an acidic medium under heating [70], by the colorimetric method of Fiske and Subbarow [70]. The values obtained were converted to μg TMP/cm^2^. To determine the amount of ionic silver (Ag^+^) released, a specific 9616 BNWP electrode (Thermo Scientific, Beverly, MA, USA) was connected to an ion analyzer (Orion 720 A^+^, Thermo Scientific, Beverly, MA, USA) and calibrated previously with standards containing 6.25, 12.5, 25, 50, and 100 μg Ag/mL. Into 1 mL of the sample, 0.020 mL of ISA (ISA, Cat. No. 940011, Thermo Scientific) was added. Values were obtained in mV, in duplicate, and converted to μg Ag/cm^2^ [71]. The data were not normally distributed and were subjected to the Kruskal–Wallis test followed by the Student–Newman–Keuls post hoc test (*p* < 0.05; SigmaPlot 12.0 software). 

### 4.7. Evaluation of the Antimicrobial Activity of Membranes

#### 4.7.1. Microorganism Strains and Growing Conditions

For this study, two strains from the American Type Culture Collection (ATCC) were used: *S. mutans* (ATCC 25175) and *C. albicans* (ATCC 10231). For *C. albicans*, colonies previously grown on Sabouraud Dextrose Agar (SDA; Difco, Le Pont de Claix, France) were used for previously grown colonies of *C. albicans* and then suspended in 10 mL of Sabouraud dextrose broth (Difco) and incubated aerobically overnight at 120 rpm and 37 °C. Simultaneously, on Brain–Heart Infusion agar (BHI Agar; Difco), previously cultured *S. mutans* colonies were suspended in 10 mL of BHI broth (Difco) and incubated overnight in 5% CO_2_ at 37 °C, statically. Subsequently, by centrifugation (8000 rpm, 5 min), bacterial and fungal cells were recovered, and their sediments were washed twice with 10 mL of 0.85% NaCl. For *C. albicans*, the cell number was adjusted to 1 × 10^7^ cells mL^−1^ using a Neubauer counting chamber and for *S. mutans*, it was adjusted using the spectrophotometer (640 nm) to 1 × 10^8^ cells mL^−1^ [72].

#### 4.7.2. Determination of Inhibition Halo

The agar diffusion method was performed with modifications [73] following the standards of the National Committee for Clinical and Laboratory Standards (*Performance Standards for Antimicrobial Disc Susceptibility Tests; Approved Standard—Eighth Edition.* Document NCCLS 13 M2-A8, 2003a). Strains of *C. albicans* (ATCC 10231) and *S. mutans* (ATCC25175) were reactivated on SDA agar (Sabouraud Dextrose, Difco, Le Pont de Claix, France) and BHI for 48 h at 37 °C in aerobiosis and microaerophilia, respectively, for each microorganism. Afterwards, colonies of each strain were inserted into BHI broth individually and incubated for 18–24 h at 37 °C. An aliquot of 300 μL of each fungal and bacterial suspension (optical density of 0.6 and absorption of 550 nm) was inserted and homogenized with 15 mL of BHI agar at 45 °C. After gelling of the culture medium, disks of 5 mm in diameter of the membranes, previously sterilized by ultraviolet radiation, were placed on the surface of the agar medium. For this experiment, a 0.2% chlorhexidine gluconate (CHX) solution was used as a positive control. To allow the solutions to diffuse, the plates were left for 2 h at room temperature and then incubated for 24 h at 37 °C. For statistical analysis, two measurements of each inhibition zone (mm) were measured with the aid of a digital caliper and the averages were calculated. The normality of the data was checked by the Shapiro–Wilk test, followed by two-way ANOVA (time and groups as factors of variation) and Fisher’s LSD post hoc test (*p* < 0.05; Software SigmaPlot 12.0).

#### 4.7.3. Cell Viability Determination

The strains of *C. albicans* (ATCC 10231) and *S. mutans* (ATCC 25175) were reactivated in SDA agar and BHI (Difco) for 48 h at 37 °C in aerobiosis and microaerophilia, respectively, for each microorganism. Cells were washed in saline solution (0.85% NaCl) and resuspended in BHI after growing overnight at 37 °C. Then, they were adjusted to a concentration of 2 × 10^7^ cells/mL for *C. albicans* and 2 × 10^8^ cells/mL for *S. mutans*. Then, three membranes from each group (5 mm) were suspended in 2 mL microtubes, containing 200 μL of cell suspensions of *S. mutans* and *C. albicans*, for 1, 2, 4, 6, 12, 18, and 24 h. After each period, the membranes were removed and resuspended in falcon tubes containing 1 mL of NaCl (0.85%). Soon after, the membranes were removed to dilute the suspension in geometric progression. Dilutions were plated on CHROMagar Candida (Difco) for *C. albicans* counting, and on BHI agar supplemented with amphotericin B (7 μg/mL; Sigma-Aldrich) for *S. mutans* counting. The agar plates were incubated at 37 °C for 24–48 h, and the number of colony-forming units (CFUs) was expressed in log10 CFU/cm² [17]. The test was performed in triplicate and on three different occasions. The normality of the data was checked by the Shapiro–Wilk test, followed by two-way ANOVA (time and groups as factors of variation) and Fisher’s LSD post hoc test (*p* < 0.05; Software SigmaPlot 12.0 software).

### 4.8. Evaluation of the Cytotoxic Activity of PA6 and TMP-AgNP Nanocompounds

#### Evaluation of Cytotoxicity by MTT Assay

Fibroblast cells of the L3T3 lineage were cultivated in DMEM culture medium, with supplementation of 10% fetal bovine serum (FBS), penicillin G (100 U/mL), streptomycin (100 μg/mL), and amphotericin B (25 μg/mL) and incubated in an oven with 5% CO_2_ at 37 °C. Every 5–7 days, the cells were subcultured, using 0.9% saline solution to wash them and 0.25% trypsin to disaggregate them from the flask. After disaggregation, the cells were centrifuged at 3000 rpm for 5 min at 15 °C, resuspended in complete DMEM (supplemented with SBF). Culture media and other reagents used in cell cultivation were purchased from Gibco^®^, Paisley, Scotland, United Kingdom. Fibroblasts, subcultured from the 3rd to 8th passages, were inoculated into 96-well microplates at a density of 1 × 10^4^ cells/well. Then, the P6 and TMP nanocompounds with and without AgNPs were added to the plate wells containing the cells and incubated in an oven at 37 °C, with 5% CO_2_. Cell viability was assessed for 24, 48, and 72 h by the 3-(4,5-dimethylthiazol-2-yl)-2,5-diphenyltetrazolium bromide (MTT) assay. For this purpose, the culture medium with the nanocompounds was removed and 10 μL of MTT solution (0.5 mg/mL) in DMEM without SBF (1:10) was added to each well. After 4 h of incubation, the MTT solution was removed, and the formazan crystals were dissolved in 100 μL of isopropyl alcohol. The plate was left in a dark chamber for 30 min on a rotary shaker at room temperature. The absorbance of the plates was evaluated at 550 nm using a microplate reader (Eon Microplate Spectrophotometer; Bio Tek, Winooski, VT, USA). All assays were performed in triplicate. The experiment was carried out in triplicate at three different times. The normality of the data was checked using the Shapiro–Wilk test, followed by two-way ANOVA (time and groups as factors of variation), followed by the Student–Newman–Keuls test (*p* < 0.05; SigmaPlot 12.0 software).

## 5. Conclusions

The synthesis of nanocompounds obtained by electrospinning proved to be stable with potential application for dental biomaterials. The antimicrobial efficacy of the nanocompounds was observed against *S. mutans* and *C. albicans*, when associated with TMP in higher concentrations, thus improving the antimicrobial effect of AgNPs. The groups with AgNPs are cytotoxic for fibroblasts and the groups with TMP nanoparticles showed less cytotoxicity, proving to be an interesting compound for the development of new biomaterials.

## Figures and Tables

**Figure 1 antibiotics-13-00340-f001:**
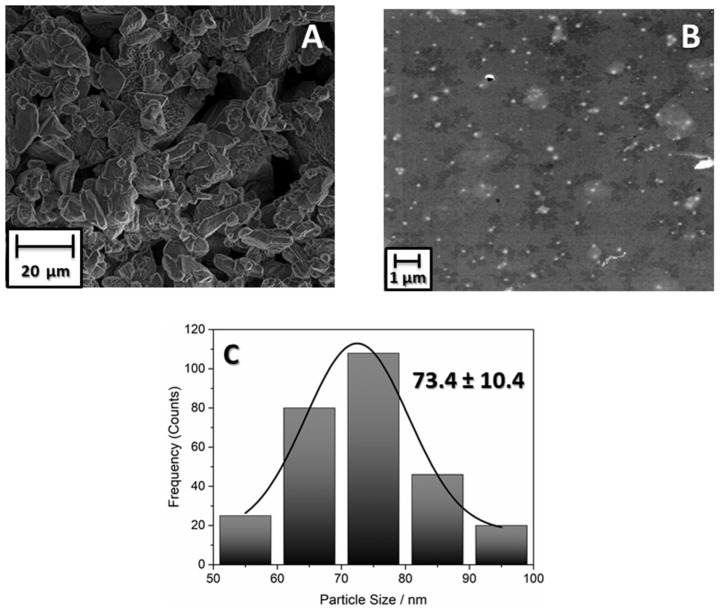
(**A**) Micrograph of TMP before ball-milling process. (**B**) Micrograph of TMP after 48 h of ball milling. (**C**) Histogram with the size distribution of TMP after ball-milling process.

**Figure 2 antibiotics-13-00340-f002:**
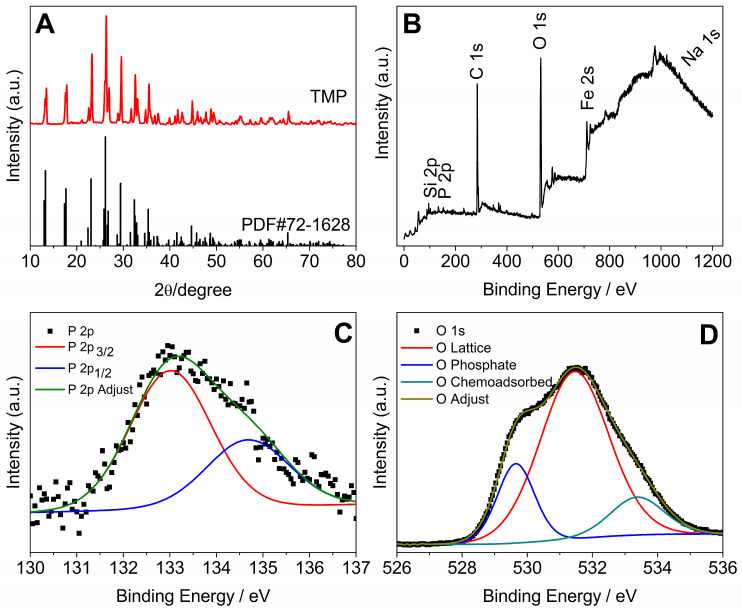
(**A**) X-ray diffractogram of TMP after milling for 48 h. (**B**) XPS survey spectrum of the TMP sample milled for 48 h. (**C**) XPS spectrum at high resolution (P 2p). (**D**) High-resolution XPS spectrum (O 1s).

**Figure 3 antibiotics-13-00340-f003:**
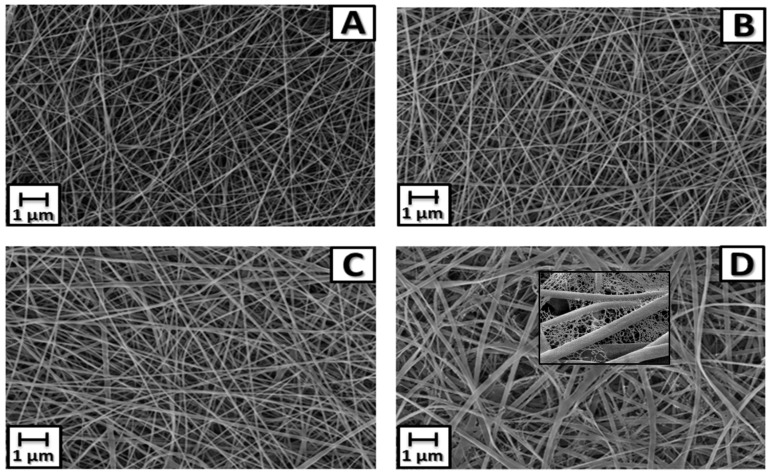
SEM images of P6 fibers and their nanocompounds. (**A**) P6 fiber. (**B**) Nanocompound P6-TMP-2.5%. (**C**) Nanocompound P6-TMP-5%. (**D**) Nanocompound P6-TMP-10%. Featured in (**D**): spider-net.

**Figure 4 antibiotics-13-00340-f004:**
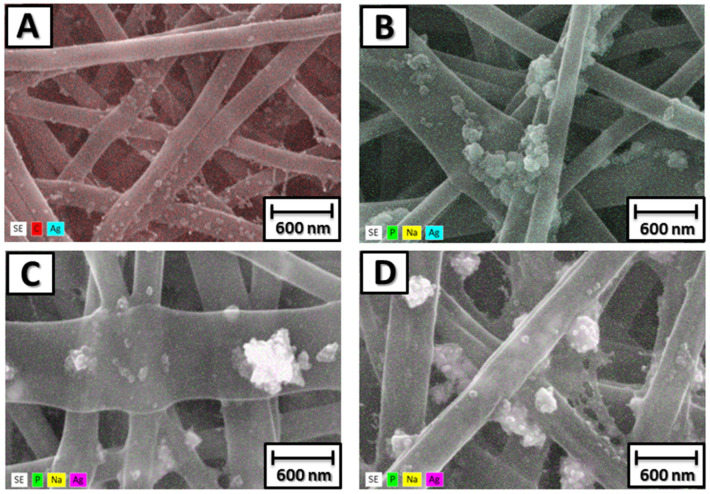
SEM images with 3D mapping of P6-TMP-AgNP fibers and their nanocompound decorated with AgNPs. (**A**) P6-Ag fiber. (**B**) Nanocompound P6-TMP-2.5%-Ag. (**C**) P6-TMP-5%-Ag nanocompound. (**D**) P6-TMP-10%-Ag nanocompound.

**Figure 5 antibiotics-13-00340-f005:**
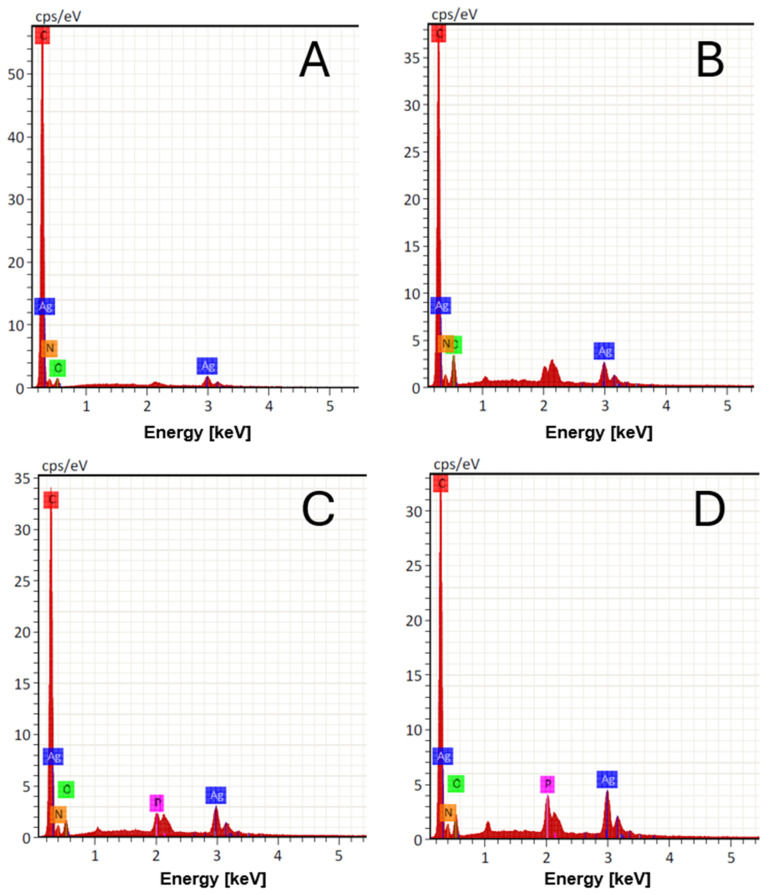
EDX histograms of P6 fibers and their nanocompounds. (**A**) P6 fiber. (**B**) Nanocompound P6-TMP-2.5%. (**C**) P6-TMP-5% nanocompound; (**D**) P6-TMP-10% nanocompound.

**Figure 6 antibiotics-13-00340-f006:**
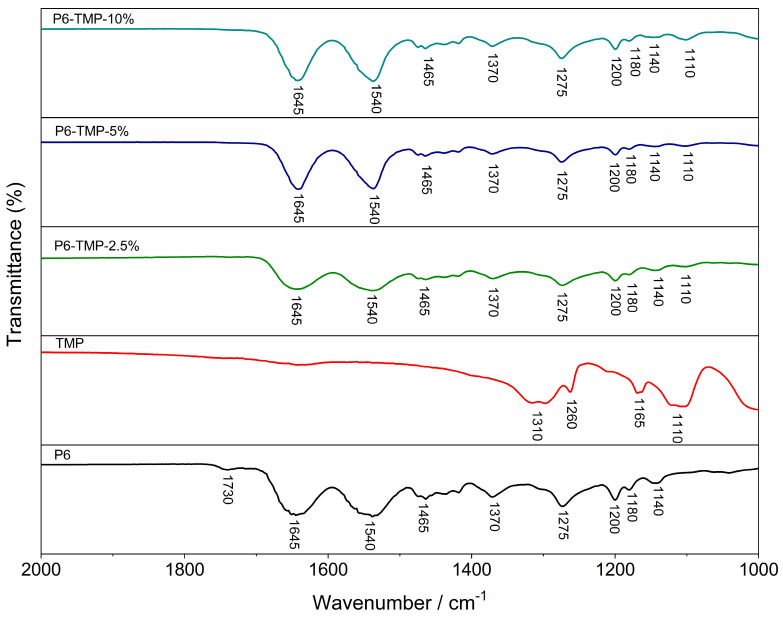
FTIR spectra of P6, TMP, and their nanocompounds.

**Figure 7 antibiotics-13-00340-f007:**
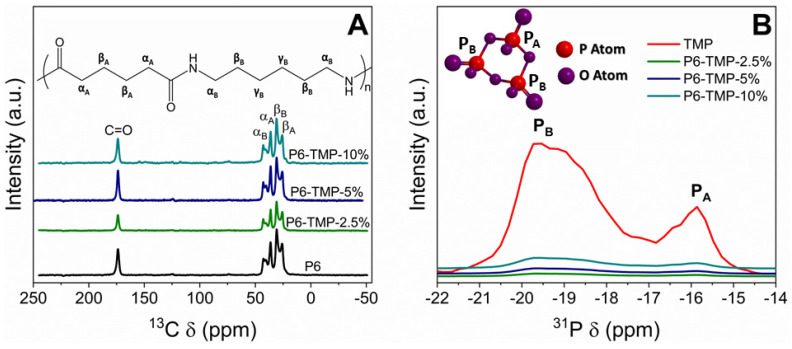
(**A**) Solid-state ^13^C NMR spectra of P6 and its nanocompounds. (**B**) Solid-state ^31^P NMR spectrum of TMP and P6-TMP nanocompounds.

**Figure 8 antibiotics-13-00340-f008:**
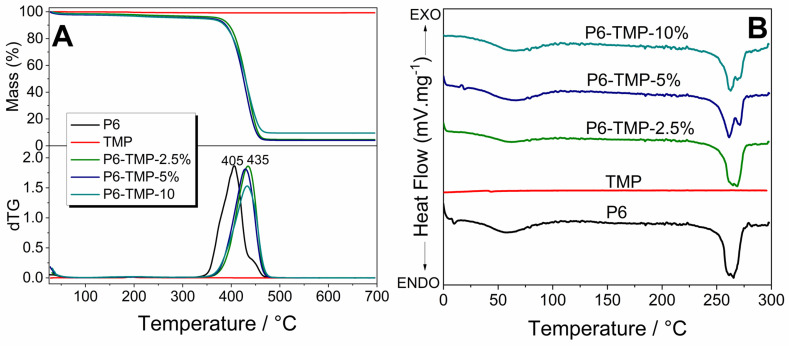
(**A**) Thermogravimetric curves and their derivatives of P6 and its nanocompounds. (**B**) DSC curves of P6 and its nanocompounds.

**Figure 9 antibiotics-13-00340-f009:**
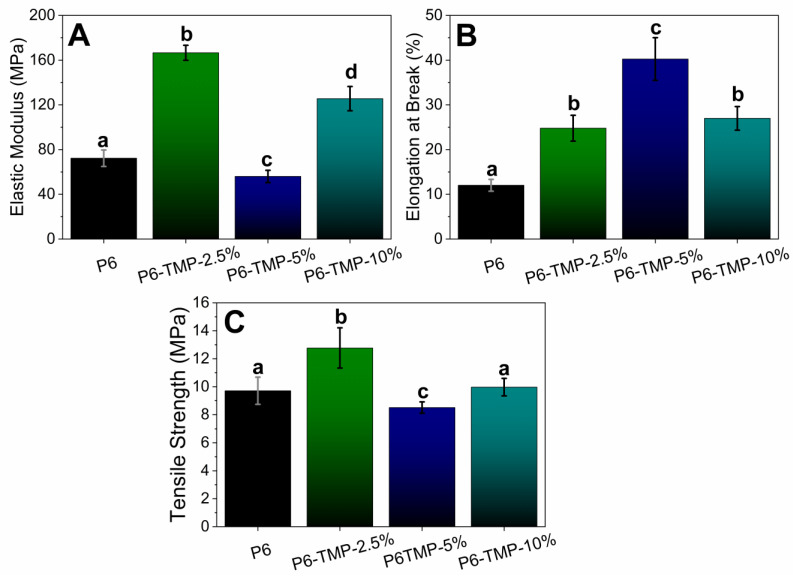
Mechanical behavior of nanocompounds. Mechanical behavior of nanocomposites. (**A**) Represents the modulus of elasticity (MPa); (**B**) Elongation at Break (%) and (**C**) Tensile Strength (MPa). Lowercase letters show statistical difference between groups (*p* < 0.05).

**Figure 10 antibiotics-13-00340-f010:**
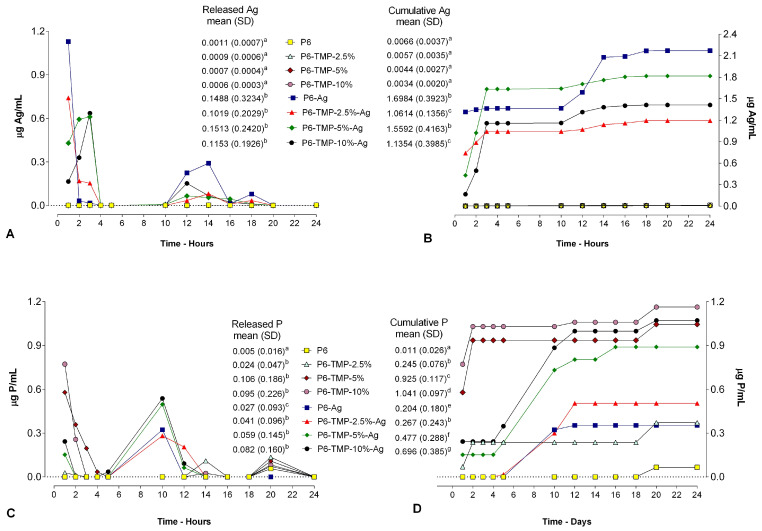
(**A**) Average Ag^+^ release values within 12 h of cycling. (**B**) Ag^+^ ionic cumulative mean values during cycling. (**C**) Mean values of TMP released within 12 h of cycling. (**D**) Cumulative mean values of TMP released during cycling. Lowercase letters indicate significant differences between means for each variable.

**Figure 11 antibiotics-13-00340-f011:**
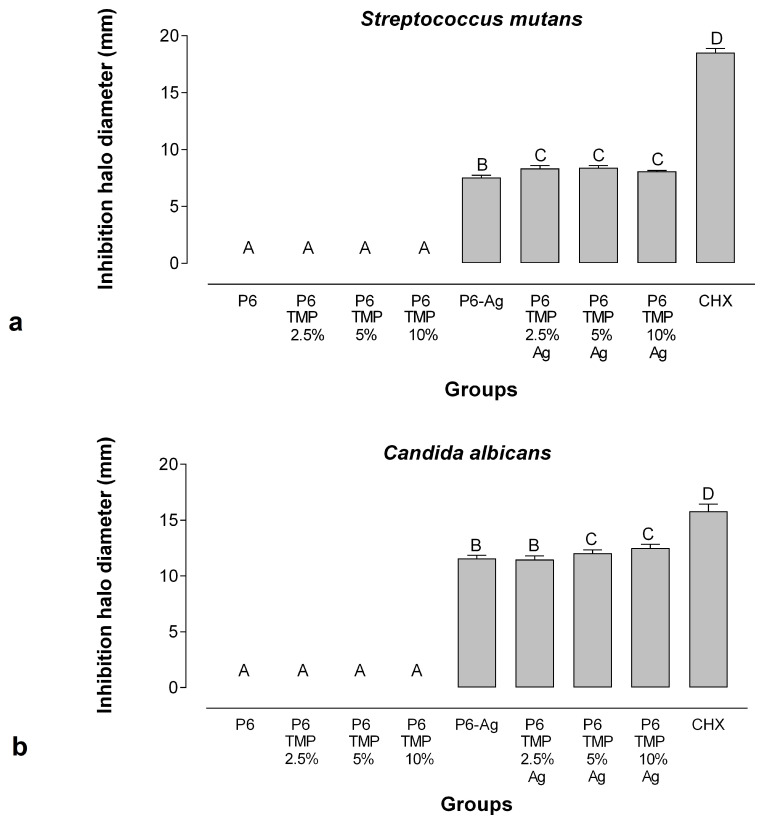
Means and standard deviation (bars) of the inhibition halo diameters for *Streptococcus mutans* (**a**) and *Candida albicans* (**b**). Different letters show statistical difference between groups.

**Figure 12 antibiotics-13-00340-f012:**
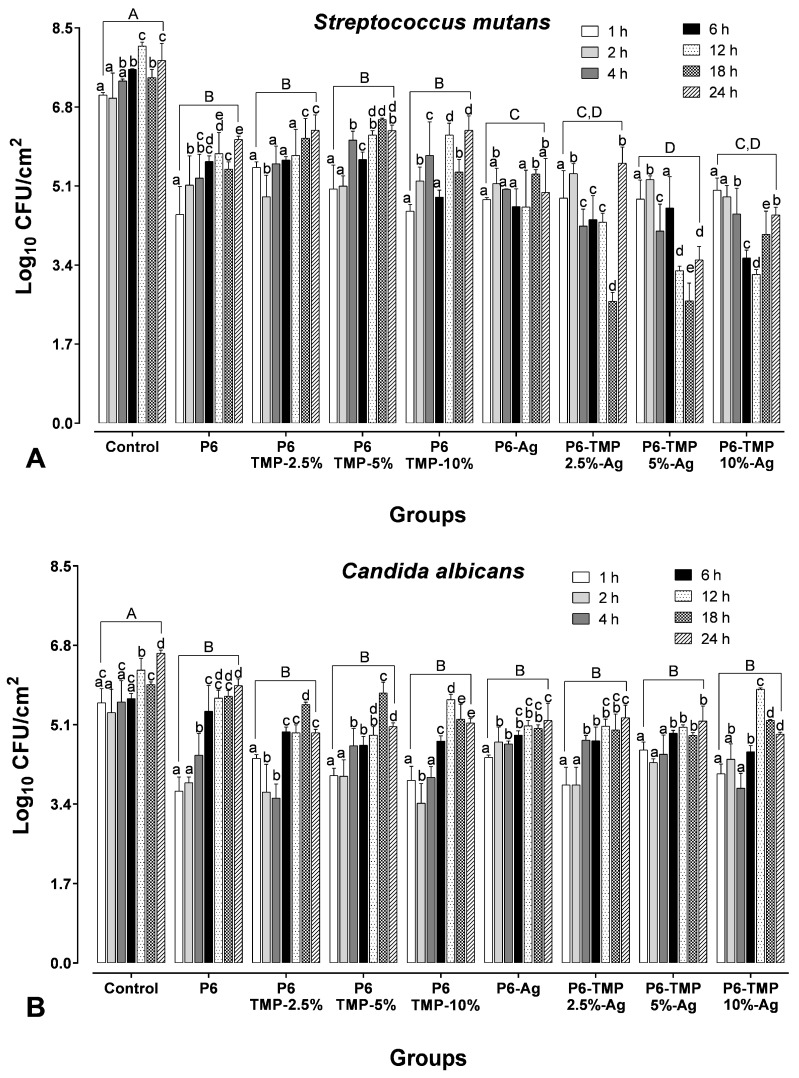
Logarithm (mean ± SD) of colony-forming units per cm^2^ for *S. mutans* (**A**) and *Candida albicans* (**B**) and in biofilms of two species. Lowercase letters show statistical difference between groups over the same period of time (*p* < 0.05). Distinct capital letters mean statistical difference between groups at a given time.

**Figure 13 antibiotics-13-00340-f013:**
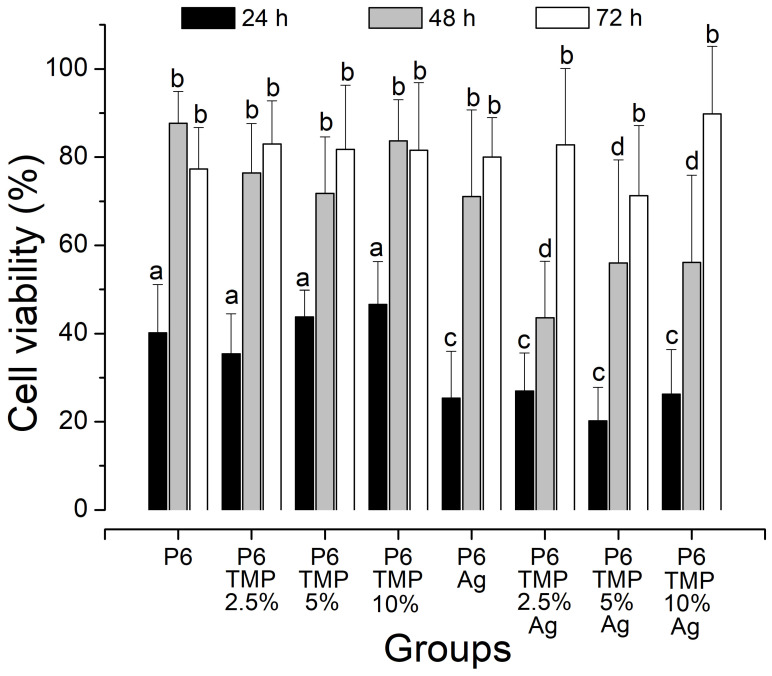
Viability of NIH/3T3 fibroblasts determined by MTT assay. Lowercase letters: comparison between the percentages of each group. Different bar colors indicate difference between times.

**Table 1 antibiotics-13-00340-t001:** MIC, MBC, and MFC values of AgNPs with NH_3_ or AgNPs without NH_3_ against the tested strains (*n* = 9).

Species	AgNPs with NH_3_		AgNPs without NH_3_	
	MIC (mg/mL)	MFC/MBC (mg/mL)	MIC (mg/mL)	MFC/MBC (mg/mL)
*C. albicans* ATCC 10231	9.40	300.94	9.40	300.94
*S. mutans* ATCC 25175	601.88	601.88	300.94	300.94

**Table 2 antibiotics-13-00340-t002:** Mean values (SD) of surface free energy (γ_s_), apolar energy (γ_s_^LW^), and polar energy (γ_s_^AB^) of P6 nanofibers with or without TMP and/or Ag nanoparticles (*n* = 7).

Nanofibers	Nano Ag	Surface Free Energy (mN/m)				
		γ_s_	γ_s_^LW^	γ_s_^AB^	γ_s_^+^	γ_s_^−^
P6	Without	41.2 ^a^ (2.2)	48.0 ^a^ (1.2)	−6.8 ^a^ (1.8)	0.6 ^a^ (0.3)	22.7 ^a^ (3.6)
P6-TMP-2.5%		28.2 ^b^ (1.6)	47.3 ^a^ (1.5)	−19.1 ^b^ (1.7)	1.8 ^b^ (0.2)	50.3 ^b^ (3.6)
P6-TMP-5%		39.2 ^a^ (1.4)	48.9 ^a^ (0.3)	−9.7 ^c^ (1.3)	0.6 ^a^ (0.2)	44.4 ^c^ (2.5)
P6-TMP-10%		40.7 ^a^ (0.9)	45.7 ^b^ (0.8)	−4.8 ^a^ (1.0)	0.2 ^c^ (0.1)	31.0 ^d^ (2.8)
P6	With	32.9 ^c^ (2.8)	50.0 ^c^ (0.4)	−17.2 ^d^ (3.0)	1.3 ^d^ (0.3)	58.9 ^e^ (5.0)
P6-TMP-2.5%		36.0 ^d^ (4.0)	48.2 ^a^ (1.2)	−12.3 ^e^ (5.0)	0.9 ^e^ (0.7)	48.0 ^b^ (4.5)
P6-TMP-5%		39.1 ^e,a^ (2.0)	48.3 ^a^ (1.2)	−9.2 ^c^ (2.9)	0.5 ^a^ (0.2)	49.1 ^f^ (4.1)
P6-TMP-10%		41.5 ^e,a^ (2.6)	45.4 ^b^ (1.7)	−3.9 ^a^ (2.4)	0.1 ^c^ (0.0)	39.0 ^g^ (3.6)

Different lowercase letters indicate a significant difference between the means for each variable (two-way Analysis of Variance, Fisher’s LSD, *p* < 0.001).

**Table 3 antibiotics-13-00340-t003:** Mean values (SD) of total-interaction free energy (ΔG_sws_^Total^) and its apolar (Δ*G*_sws_^LW^) and polar (Δ*G*_sws_^AB^) components between P6 nanofibers with or without TMP and/or Ag nanoparticles and water (*n* = 7).

Nanofibers	Nano Ag	Total Free Energy of Interaction (mN/m)		
		Δ*G*_sws_^Total^	Δ*G*_sws_^LW^	Δ*G*_sws_^AB^
P6	Without	−15.3 ^a^ (6.1)	−10.2 ^a.b^ (0.8)	−5.1 ^a^ (6.5)
P6-TMP-2.5%		20.4 ^b^ (3.4)	−9.7 ^b^ (0.9)	30.2 ^b^ (3.4)
P6-TMP-5%		16.9 ^b^ (3.0)	−10.8 ^a^ (0.2)	27.7 ^b^ (3.1)
P6-TMP-10%		0.9 ^c^ (4.6)	−8.6 ^c^ (0.5)	9.5 ^c^ (4.4)
P6	With	29.5 ^d^ (3.6)	−11.6 ^d^ (0.2)	41.1 ^d^ (3.7)
P6-TMP-2.5%		20.8 ^b,e^ (5.6)	−10.4 ^b^ (0.8)	31.1 ^b,e^ (5.5)
P6-TMP-5%		23.7 ^e^ (4.2)	−10.4 ^a,b^ (0.8)	34.1 ^e^ (4.3)
P6-TMP-10%		13.9 ^f^ (5.2)	−8.6 ^c^ (1.1)	22.5 ^f^ (4.7)

Lowercase letters indicate significant differences between means for each variable (two-way Analysis of Variance, Fisher’s LSD, *p* < 0.001).

## Data Availability

The data presented in this study are available on request from the corresponding author.

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
