# Peer review of "Synthesis, Characterization, and Evaluation of the Antimicrobial Effects and Cytotoxicity of a Novel Nanocomposite Based on Polyamide 6 and Trimetaphosphate Nanoparticles Decorated with Silver Nanoparticles"

_antibiotics, 2024, doi:10.3390/antibiotics13040340_

Round 1

Reviewer 1 Report

Comments and Suggestions for Authors

The manuscrispt is based on the study of the antimicrobial activity against Streptococcus mutans and Candida albicans of the polyamide-6 polymer matrix impregnated with trimetaphosphate and silver nanoparticles, the release of thses nanoparticles and also the cytotoxicity. The subject chosen by the authors is interesting and topical.

After solving some minor requirements (see below), I agree with the publication of this manuscript in Antibiotics journal.

1. Some considerations about the materials obtained by the authors, in comparison with other materials used as scaffolds for the regeneration of dental tissues (from the point of view of the obtaining method, antimicrobial efficiency, cytotoxicity, etc.) in order to emphasize the importance of the synthesized materials.

2. I have noticed some typos:

- Table 1, second line – MIC, not CMI

- line 47 – interstitial space

- check line 159

- Figure 10 - The meaning of the OX and OY axes should be written in English

- line 252 – „... death cell phone”. Please, consider changing of this phrase without "phone"

- line 404 – in English

Comments on the Quality of English Language

Some mistakes were mentioned above.

Author Response

Hi,

Thank you. 

Reviewer 2 Report

Comments and Suggestions for Authors

The abstract would benefit from providing some context to the research and omitting the data analysis method from line 25. It could also be clearer if rephrased for easier understanding.

Adding more relevant keywords on line 36, such as 'silver nanoparticles' and 'trimetaphosphate nanoparticles,' would improve search engine discoverability.

It's unclear on line 66 how this research contributes to the field and how it compares to Carla Corrêa Mendes-Gouvêa's work. Adding a brief explanation would be helpful.

More details about the TMP nanoparticle synthesis, including size, shape, and surface chemistry, would be valuable. Similarly, understanding the release kinetics of TMP and Ag+ ions at different temperatures and pHs would strengthen the study.

Using different colors for each group in Figure 9 would improve its clarity.

Comments on the Quality of English Language

Translating Figure 10 into English would make it accessible to a wider audience.

The meaning of 'death cell phone' on line 252 is unclear. Please clarify.

Additionally, translating the title of section 4.4 (line 404) into English would enhance readability.

Round 2

Reviewer 2 Report

Comments and Suggestions for Authors

The abstract is quite detailed, which is good for understanding the study, but it could be condensed for better readability. Consider summarizing some points to make it more concise. Instead of listing all the results in detail, focus on highlighting the key findings. Mentioning the most significant outcomes, such as the antimicrobial activity, release kinetics of TMP and Ag+, and cytotoxicity results, will give readers a clear understanding of the study's main contributions. While mentioning the statistical analysis is important, it might not be necessary to include specific test names like "Fisher's LSD test" in the abstract. Simply removing the statistical analysis sentence, or stating that statistical analysis was performed and indicating the significance level would suffice. The conclusion provides a good summary of the study's findings. However, you could emphasize the practical implications of the results and the potential significance for dental tissue scaffolding applications. For example:"Overall, the developed P6 polymeric matrix impregnated with TMP and AgNPs demonstrated promising antimicrobial properties against tested microorganisms, suggesting its potential utility in dental tissue scaffolding applications."
In the Materials and Methods section, please provide more details on the statistical analysis methods used to analyze the data. 

Comments on the Quality of English Language

In the abstract, I'd recommend including an object when you're using a comparative adjective to make the sentence clearer. For example, "The groups with P6-TMP showed higher cell viability" than xxx (the groups with xxx).

Please avoid run-on sentences by breaking them down into smaller, more digestible parts might improve readability. For example, the first sentence in the introduction (line 35-37) can be revised as "The loss of tissue resulting from dental caries or trauma underscores the importance of research in tissue engineering. This field focuses on developing biomaterials aimed at regenerating tissue structure and restoring both form and function."

Is "nuclear" (line 47) a typo?

For all the examples provided above, please utilize your discretion to revise them as needed. Additionally, please keep in mind that these examples represent only a small subset of similar cases; therefore, revisions should be made comprehensively across all relevant instances.

Author Response

Hi, 

Thank you. 
